# Cognitive Impairment in Multiple Sclerosis: An Update on Assessment and Management

**Emilio Portaccio** [1],* and **Maria Pia Amato** [1,2]

1   Department of Neurofarba, University of Florence, 50139 Florence, Italy
2   IRCCS Fondazione Don Carlo Gnocchi, Department of Neurology, 50143 Florence, Italy
*   Correspondence: emilio.portaccio@unifi.it

**Abstract:** Cognitive impairment (CI) is a core feature of multiple sclerosis (MS) and affects up to 65% of patients in every phase of the disease, having a deep impact on all aspects of patients' lives. Cognitive functions most frequently involved include information processing speed, learning and memory, visuospatial abilities, and executive function. The precise pathogenetic mechanisms underpinning CI in MS are still largely unknown, but are deemed to be mainly related to pathological changes in lesioned and normal-appearing white matter, specific neuronal grey matter structures, and immunological alterations, with particular impact on synaptic transmission and plasticity. Moreover, much research is needed on therapeutic strategies. Small to moderate efficacy has been reported for disease-modifying therapies, particularly high-efficacy drugs, and symptomatic therapies (dalfampridine), while the strongest benefit emerged after cognitive training. The present narrative review provides a concise, updated overview of more recent evidence on the prevalence, profile, pathogenetic mechanisms, and treatment of CI in people with MS. CI should be screened on a regular basis as part of routine clinical assessments, and brief tools are now widely available (such as the Symbol Digit Modalities Test). The main goal of cognitive assessment in MS is the prompt implementation of preventive and treatment interventions.

**Keywords:** multiple sclerosis; cognitive impairment; information processing speed; memory; depression; fatigue; screening; cognitive rehabilitation





## 1. Introduction

Multiple sclerosis (MS) is a chronic, inflammatory, demyelinating disease of the CNS with a significant healthcare burden for the patient, their family, and the community [1]. The typical onset of MS is during young adulthood and is mainly characterized by recurrent episodes of neurological dysfunction from which the individual usually recovers (the relapsing–remitting (RR) course) [1]. Over time, inflammatory activity reduces, neurological disability worsens independently of relapses, and the disease enters the progressive course (the secondary progressive course). In a small proportion of subjects (approximately 10%), the MS course is progressive from onset (primary progressive MS) [1]. This dichotomous view of the disease has recently been challenged, since neurodegeneration and clinical progression independent of relapse activity can be detected even in the earliest "inflammatory" phases of the disease, indicating MS as a single continuum in which inflammation and neurodegeneration coexist since onset [2–4]. Despite the historical description of "enfeeblement of memory" and "slow concept formation" by Charcot [5], neuropsychological dysfunction was overlooked for a long time in patients with MS. However, in the past three decades, cognitive impairment (CI) has received increasing attention and investigation, and it is now widely acknowledged as a core feature of MS, negatively impacting physical independence and competence in activities of daily living [6,7]. In this narrative review, we provide a brief, updated overview of CI in MS, mainly focusing on its prevalence and neuropsychological profile, potential pathogenetic mechanisms, and treatment opportunities.

## 2. Prevalence and Profile of Cognitive Impairment in Multiple Sclerosis

CI is highly prevalent in MS, affecting 34–65% of patients during the course of the disease [6,7]. Cognitive deficits can occur in every stage of MS, even in the earliest "pre-clinical" phase, the radiologically isolated syndromes, in which the disease manifests only on magnetic resonance imaging (MRI) and laboratory examinations [6,7]. CI can progress insidiously and gradually, or abruptly, during relapses; in the past few years, isolated cognitive relapses with an exclusive involvement of cognitive performance have been described [7,8]. Overall, the frequency and severity of CI tend to increase over time and become more pronounced in the progressive courses [6,7]. It has been estimated that the rate of cognitive dysfunction is approximately 20–25% in clinically and radiologically isolated syndromes, 30–45% in RRMS, and 50–75% in secondary progressive MS [6,7]. However, it has been demonstrated that the main factors associated with CI are greater physical disability, as measured by the Expanded Disability Status Scale (EDSS), and older patients' age, rather than longer disease duration or the MS course per se [9]. The presence of CI has been associated with a worse prognosis: it increases the risk of conversion from clinically isolated syndromes to clinically definite MS [10], the risk of disability progression over time [11,12], as well as the risk of death [13]. The association with mortality was attributed to the relationship between cognitive dysfunction and more widespread neuropathology [13]. Moreover, CI impacts participation in social activities, driving abilities, and employment status and overall reduces the health-related quality of life [6,7]. Therefore, given its clinical and prognostic relevance, routinely assessment of CI is of critical importance for a comprehensive evaluation of MS patients. At the individual level, the profile of CI varies widely, as all cognitive domains can be affected. However, at the group level, when clustering of neuropsychological impairment in different domains is the focus of the investigation, information processing speed, learning and memory, visuospatial abilities, and executive functions are more frequently involved [6,7]. On the other hand, when a pattern of cognitive deficits in a single patient is assessed, a different cognitive profile can arise. Recently, in a large sample of 1212 MS patients, five phenotypes of cognitive functioning have been identified through a latent profile analysis: preserved cognition (detected in 19.4% of subjects), mild verbal memory/semantic fluency involvement (detected in 29.9%), mild multidomain involvement (detected in 19.5%), severe-executive/attention involvement (detected in 13.8%), and severe multidomain impairment (detected in 17.5%) (Figure 1) [14].

Beyond the cognitive domains described above, other cognitive functions and processes can be involved in MS. Recent observations reported differential impairment of the core aspects of social cognitive processing in patients with MS [15]. Moreover, alterations of learning and memory processes, together with typical dysfunctional behaviours, such as deficits in action control and motor inhibition, have been found to be core factors in different neurodegenerative disorders [16]. Other aspects, relatively less evaluated in MS, such as altered emotion perception, can contribute to cognitive dysfunction [17].

General intelligence and language, generally spared in adult-onset MS patients, can be impaired in paediatric onset MS (POMS), in which the disease manifests before the age of 18 years [18]. In this age range, MS-related brain damage occurs during the formative years and interferes with normal neuronal maturation and the development of cognitive reserve. On the other hand, POMS subjects may have higher repair capabilities, possibly due to their higher neuroplasticity. Indeed, CI is detectable in approximately one third of people with POMS and can have a heterogeneous course over time, with an overall tendency toward a recovery at the group level [18]. However, a recent 12-year follow-up observational study revealed the proportion of patients with impairment at the last evaluation was more than double that at baseline [19]. Moreover, worse cognitive performances were associated with lower psychosocial attainment in adulthood [19]. On the other hand, neuropsychological involvement holds some peculiarities in late-onset MS patients (LOMS), in which the disease manifests after the age of 50 years. LOMS shows more frequent impairment in visual learning and memory, working memory, and semantic fluency tests [20,21],

although, on the whole, some evidence suggests that the neuropsychological profile can be comparable to that of the general MS population [7,22]. Moreover, in these patients, MS-related cognitive impairment poses diagnostic challenges since it should be differentiated from other causes of CI, such as Alzheimer's disease and vascular CI. A comprehensive diagnostic work-up, including extensive neuropsychological examinations and specific laboratory (cerebrospinal fluid biomarkers) and radiological testing (amyloid PET), might be needed in older MS subjects with impending cognitive dysfunction.

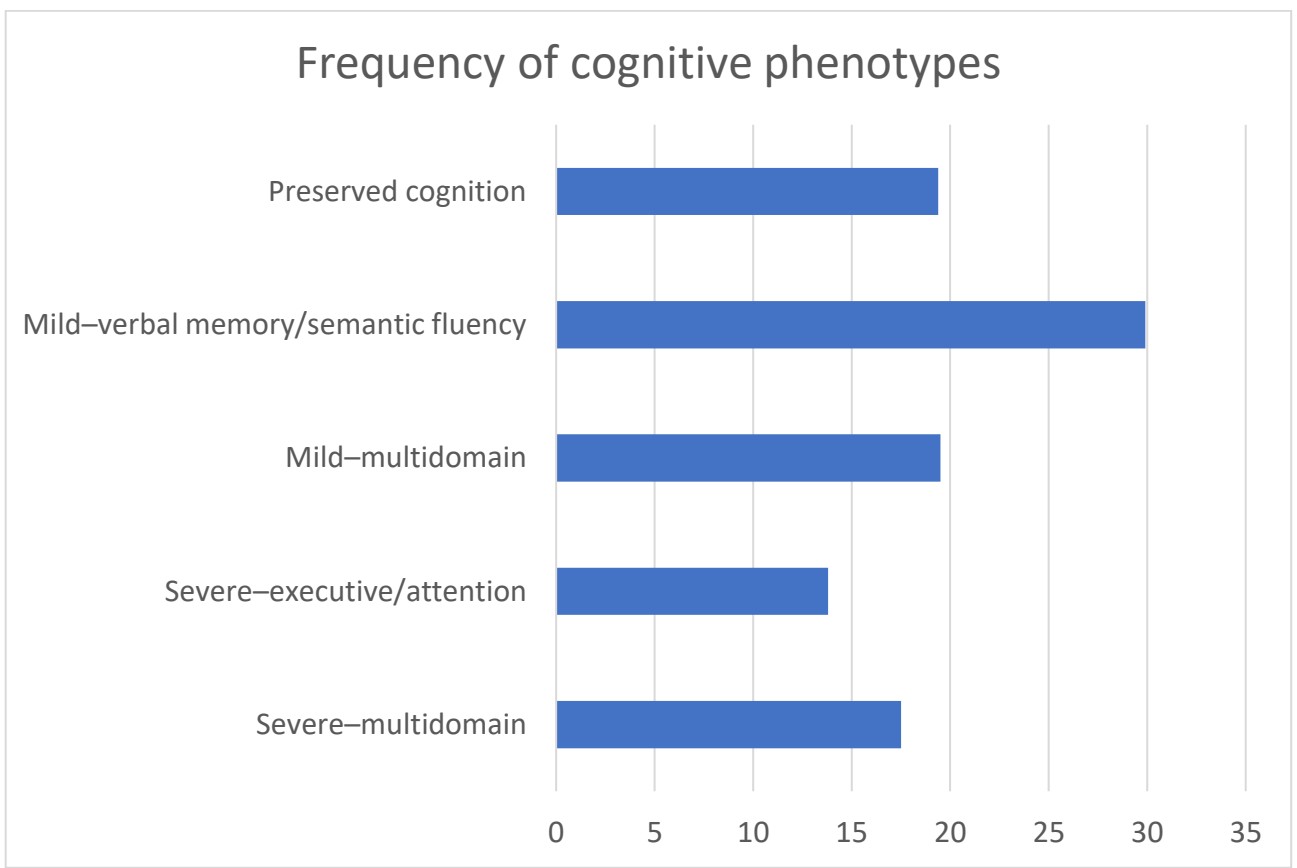

**Figure 1.** Percentage of different cognitive phenotypes in multiple sclerosis patients [14].

Several factors can impact cognition and should be considered in the assessment and management of neuropsychological changes in patients with MS. Among those, depression, anxiety, fatigue, and sleep disorders have been studied more extensively. Primary failure of key brain regions involved in emotional processing and regulation or abnormal connectivity between them can contribute to the adoption of maladaptive cognitive strategies and the development of mood disorders. In particular, depression could negatively affect working memory and, more specifically, executive control [23–25]. Likewise, fatigue and sleep disorders have been linked to deficits in processing speed, memory, attention, and executive functions [26–31].

### 3. Pathogenesis of Cognitive Impairment in Multiple Sclerosis

The precise mechanisms underpinning CI in MS are still largely unknown. Overall, it is argued that the pathogenesis of MS-related CI is multifactorial. Pathological changes in lesioned and normal-appearing CNS white matter and specific neuronal grey matter structures could play a crucial role [6,7,32], along with alterations in the physiological crosstalk between the immune and nervous systems, with particular impact on synaptic transmission and plasticity [32,33]. While several factors hinder the identification of specific CNS structures and/or circuits related to specific cognitive domains, it is possible

to speculate that widespread focal white matter lesions can be mainly related to the impairment of information processing speed, suggesting MS-related CI as a disconnection syndrome [34–36]. In early MRI studies, white matter lesions on T2-weighted scans partially correlated with the severity of CI in MS [37]. More recent studies based on more advanced MRI techniques, such as magnetisation transfer, diffusion tensor imaging, T1 relaxometry, and double inversion recovery, identified more subtle and widespread brain damage robustly correlated with cognitive decline [7,38–43].

Moreover, recent evidence indicates that thalamic damage may contribute to the disruption of cortico-subcortical and cortico-cortical connections [44]. On the other hand, the involvement of cortical areas can account for failure in specific cognitive domains; such failure occurs for learning and memory in cases of hippocampal damage or for executive functions in cases of frontal lobe alterations [6,7,32]. Both grey matter volumes and focal cortical damage (cortical lesions) have been linked to cognitive impairment in MS [41,42,45–48]. Furthermore, diffuse, immune-mediated functional alteration of synaptic GABAergic and glutamatergic transmission contributes to the disruption of neuronal network functioning [32,33,49]. Interestingly, combining blood and imaging measures using cross-modal biomarkers can improve the accuracy of predicting CI in MS [50,51].

Other pathogenetic mechanisms, less assessed in MS-related cognitive dysfunction, such as alterations in metabolic pathways and mitochondrial resilience, associated with various psychiatric, neurological, and neurodegenerative illnesses, can contribute to CI, with a particular impact on social cognition and social functioning [52–54].

In a more integrated view, all the above-mentioned mechanisms work synergistically to cause disruptions of structural and functional connections that are the basis of normal brain functioning. Damage to brain regions (i.e., the nodes of the network) as well as their anatomical (i.e., edges of the network) and functional connections progressively reduces network efficiency until a "network collapse", which is tied to faster neurodegeneration and accelerated clinical and cognitive deterioration [55].

## 4. Neuropsychological Assessment

Since the seminal paper by Rao and colleagues [56], different cognitive batteries for MS have been developed, mainly tapping domains deemed to be disease-specific. Therefore, tests of processing speed, attention, verbal, and spatial memory have been more frequently included. Although lengthy and extensive neuropsychological batteries have been abandoned, the need for even 15 min of one-on-one testing for every patient could not be practical, hindering regular cognitive monitoring in MS standard care. The Symbol Digit Modalities Test (SDMT) showed higher sensitivity for cognitive dysfunction in MS and is now widely acknowledged as the gold standard for a quick cognitive screening [57–59]. However, the test is not specific and is limited to processing speed assessment, overlooking other relevant cognitive domains such as learning and memory (see cognitive profile section). Therefore, a more comprehensive assessment with referral to a neuropsychologist should always be considered in cases of failure on cognitive screening as well as in cases of high suspicion of CI and normal performance on the screening tool [59]. The main brief cognitive batteries for MS are reported in Table 1. Computerized testing can represent a valid alternative to conventional paper-and-pencil assessment in order to address time constraints and increase the feasibility of routine cognitive screening. Moreover, the automation of computer-mediated tests can minimise the need for trained professionals/technicians. To date, the validity of some computer-assisted tests has been established in patients with multiple sclerosis [60], such as the Processing Speed Test (PST) [61], a tablet-based test modelled after the SDMT.

**Table 1.** Main validated tests for cognitive impairment in multiple sclerosis.

| | Duration of Administration | Tests Included | Cognitive Functions Assessed |
|---|---|---|---|
| **Screening tests** | | | |
| Symbol Digit Modalities Test (SDMT) [57] | 5 min | Symbol Digit Modalities Test (SDMT) | Information processing speed |
| Processing Speed Test (PST) [61] | 5 min | Processing Speed Test (PST) | Information processing speed |
| Computerized Speed Cognitive Test (CSCT) [62] | 5 min | Computerized Speed Cognitive Test (CSCT) | Information processing speed |
| **Brief Neuropsychological Batteries** | | | |
| Brief International Cognitive Assessment for Multiple Sclerosis (BICAMS) [58] | 15 min | Symbol Digit Modalities Test (SDMT) California Verbal Learning Test–2nd ed. (CVLT-II) Brief Visuospatial Memory Test–Revised (BVMTR) | Information processing speed Verbal learning and memory Visuospatial learning and memory |
| Brief Repeatable Neuropsychological Batter (BRNB) [56] | 45 min | Paced Auditory Serial Addition Test (PASAT) Symbol Digit Modalities Test (SDMT) Selective Reminding Test (SRT) 10/36 Spatial Recall Test (SPART) Controlled Oral Word Association Test (COWAT) | Attention Information processing speed Verbal learning and memory Visuospatial learning and memory Verbal fluency |
| Minimal Assessment of Cognitive Function in Multiple Sclerosis (MACFIMS) [63] | 90 min | Paced Auditory Serial Addition Test (PASAT) Symbol Digit Modalities Test (SDMT) California Verbal Learning Test–2nd ed. (CVLT-II) Brief Visuospatial Memory Test–Revised (BVMTR) Controlled Oral Word Association Test (COWAT) Judgement of Line Orientation Test (JLoT) Delis-Kaplan Executive Function System Sorting Test (D-KEFS-ST) | Attention Information processing speed Verbal learning and memory Visuospatial learning and memory Verbal fluency Visuospatial perception Executive functions |

Overall, according to recent recommendations, cognitive screening should be performed at the first evaluation and repeated annually with the same instrument or more often as needed [59]. In cases of cognitive failure at screening evaluation, a more comprehensive assessment should be planned [59]. For children, cognitive evaluation should also include monitoring of academic and behavioural school functioning [59].

Since mood, fatigue, and sleep disorders are widely acknowledged as important contributors to CI in MS, a comprehensive neuropsychological assessment should always include routine monitoring and screening of these factors for a formulation of the patient's psychological state and difficulties; the Beck Depression Inventory—Fast Screen [64] or the Hospital Anxiety and Depression Scale [65] for adults, or an age-appropriate screening tool for children, have been recently recommended [59].

## 5. Treatment of Cognitive Impairment in Multiple Sclerosis

Despite the high prevalence and deep impact on patients' lives, there are no approved medications for the treatment of CI in MS. Taking into account the pathogenetic hypotheses discussed above, in principle, disease-modifying therapies (DMTs) should improve cognition in addition to the traditional outcomes of relapse rate and disability progression. In a recent systematic review and meta-analysis collecting data from 55 cohorts from 44 studies, an overall beneficial effect of DMTs on cognition emerged, although the effect size was small to medium and the quality of the research was low for the majority of the studies [66]. Indeed, in a subsequent systematic review evaluating the effect size of pharmacological interventions on cognition, the authors did not find any significant effect [67]. It has to be noted that cognitive measures (SDMT) have been included as a tertiary or exploratory outcome in phase 3 trials only in the last few years, and further data about DMT's efficacy on cognition are expected in the future. For instance, there is recent preliminary evidence of the beneficial effect of DMTs on cognition, particularly high-efficacy DMTs (such as

sphingosine-1-phosphate modulators) [68,69]. As for symptomatic pharmacological treatment, drugs such as modafinil, donepezil, l-amphetamine sulfate, and memantine have shown conflicting effects on MS-related cognitive impairment [6]. On the other hand, a recent meta-analysis, including class I, class II, and class IV randomised controlled trials, found a positive effect of dalfampridine over placebo on SDMT scores [70]. However, the effects were generally transient, and further confirmation is needed.

Stronger evidence exists for cognitive training. Cognitive rehabilitation includes restorative approaches, which attempt to restore the impaired cognitive function (often through intensive cognitive training programs), or compensatory strategies, which exploit residual or spared cognitive functions. A meta-analysis of 20 randomised controlled trials on restorative rehabilitation found a moderate effect size among treated patients [71]. Among available computerised programmes, RehaCom was the most investigated in MS, with improvements in attention, information processing speed, memory, and executive function.

As for strategy-based compensatory approaches, the strongest evidence emerged in the rehabilitation of memory impairment through the modified Story Memory Technique. This technique trains patients to use context and imagery as strategies to improve the acquisition and retention of information; class I evidence of its efficacy was recently provided [72].

Finally, beside the interventions aimed at restoration/remediation of cognitive deficits, a comprehensive management of neuropsychological dysfunction in people with MS must address concomitant mood disorders and other factors impacting cognition. Among non-pharmacological strategies, psychological interventions such as mindfulness-based approaches, have shown to be effective in improving cognitive functioning in MS [73,74].

## 6. Conclusions and Future Directions

CI is now widely acknowledged as a prevalent core feature of the MS clinical picture (Table 2). Given its critical consequences on patients' lives, physicians should be sensitized to routinely assess and screen cognitive function in MS patients. The SDMT is a quick, easy, and sensitive measure that should become part of the standard clinical assessment of MS. However, since it focuses mainly on information processing speed, a wider cognitive screening should be considered in patients at high risk of CI, such as those with a great disease burden on MRI or who report cognitive difficulties or significant changes in work and everyday life activities. Screening for mood disorders and other factors impacting cognition, such as fatigue and sleep disorders, should be part of routine monitoring. More focused neuropsychological batteries and laboratory/radiological testing should be performed in specific conditions, such as in POMS and LOMS. The main goal of cognitive assessment in MS is the prompt implementation of treatment interventions. Unfortunately, to date, there are no approved medications for CI in MS. DMTs are deemed to improve cognition and prevent further deterioration through their effectiveness on potential contributors to neuropsychological dysfunctions (lesion load, grey and white matter atrophy, immunological alterations). It is still debated whether the worsening of cognitive functioning should lead to a modification and escalation of DMT. While preliminary evidence about the about efficacy of dalfampridine is emerging, strongest benefit has been reported after cognitive training. A better understanding of the pathogenesis of CI in MS will allow the identification of more focused and effective treatment approaches, able to restore brain network efficiency. It is conceivable that multimodal, personalized approaches, including pharmacological and non-pharmacological interventions, could achieve greater advancements in the management of CI in people with MS.

**Table 2.** Take home messages.

| |
|---|
| Cognitive impairment is now widely acknowledged as a prevalent core feature of the MS clinical picture, with critical consequences on patients' lives. |
| The Symbol Digit Modalities Test is a quick, easy, and sensitive measure that should routinely be performed in MS clinical assessments. Wider cognitive screening should be considered in patients at high risk of cognitive impairment. |
| Cognitive screening should include evaluation of mood disorders and other factors impacting cognition, such as fatigue and sleep disorders. |
| Unfortunately, to date there are no approved medications for cognitive impairment in MS. It is conceivable that multimodal, personalized approaches, including pharmacological and non-pharmacological interventions, could allow better management of MS-related cognitive impairment. |

MS: multiple sclerosis.

**Funding:** This research received no external funding.

**Conflicts of Interest:** EP received compensation for travel grants, participation in advisory board and/or speaking activities from Biogen, Merck Serono, Sanofi, Teva, and Novartis; serves on the editorial board of Frontiers in Neurology and Brain Sciences. MPA served on scientific advisory boards and has received speaker honoraria and research support from Biogen Idec, Merck Serono, Bayer Schering Pharma, and Sanofi Aventis, and serves on the editorial board of Multiple Sclerosis Journal and BMC Neurology.

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
