# Peer review of "Cognitive Impairment in Multiple Sclerosis: An Update on Assessment and Management"

_neurosci, doi:10.3390/neurosci3040048_

Round 1

Reviewer 1 Report

I looked over the paper titled "Cognitive impairment in multiple sclerosis: an update."  It is an important topic, however, I could not find any updated information.
The importance of cognitive assessments in clinical and prognostic aspects of MS could be discussed. Moreover, the effects of DMTs on cognitive symptoms and the correlations between CI and MRI, and the clinical status of MS patients could be discussed more. 

Author Response

  • We thank the reviewer for her/his suggestions. While the main objective of our review was to provide a brief and concise overview of cognitive impairment in multiple sclerosis, the paper has been expanded, as suggested

Reviewer 2 Report

Thank you for asking me to review this interesting narrative review on cognitive impairments in people with multiple sclerosis. I believe the article is relevant to the readership of NeuroSci. However, a number of major issues need to be addressed before it can be accepted for publication: 

1. While the theorised pathogenesis of cognitive impairment in MS is covered extensively from a medical perspective, particularly in terms of damage to the CNS, no mention is made of any potential psychological factors. This needs to be addressed, since the detrimental impact of psychological difficulties such as depression and anxiety on the cognitive functioning  of people with MS has been extensively studied and proven (e.g.,https://doi.org/10.1016/j.jns.2005.08.020, https://doi.org/10.1016/j.msard.2018.07.029). 

2. Similarly, the section on Neuropsychological Assessment should also mention the need for referrals to neuropsychologists not only to assess cognition, but also to provide a comprehensive formulation of patients' psychological state and difficulties. 

3. In the Conclusions, the authors state that new forms of personalised and non-pharmacological interventions should be explored in the future. However, the review only mentions cognitive rehabilitation as non-pharmacological interventions, with no discussion around forms of psychological therapy which may help improve cognitive impairments in MS. This needs to be addressed, since psychological interventions such as mindfulness-based approaches have shown to be effective at improving cognitive functioning both in MS (https://doi.org/10.1080/09638288.2022.2069292) and comparable conditions such as premanifest/early stage Huntington's disease (https://doi.org/10.3233/JHD-210471).

Minor points: 

- On line 34 the author say "despite the historical description", without specifying what such description entailed. This could be added briefly. 

- Starting on lines 52-53 it is not clear what the authors mean with "the main determinants of CI are greater physical disability..." - does this mean greater physical disability predicts higher CI or vice versa? The whole paragraph up until line 70 could use a comprehensive revision to improve clarity. It would also be helpful if the authors detailed what hypotheses/explanations have been proposed for CI to increase risk of death. 

- The whole paper would benefit from being proofread by a native speaker of English in order to address inaccuracies and improve flow and readability. 

Author Response

  1. While the theorised pathogenesis of cognitive impairment in MS is covered extensively from a medical perspective, particularly in terms of damage to the CNS, no mention is made of any potential psychological factors. This needs to be addressed, since the detrimental impact of psychological difficulties such as depression and anxiety on the cognitive functioning  of people with MS has been extensively studied and proven (e.g.,https://doi.org/10.1016/j.jns.2005.08.020, https://doi.org/10.1016/j.msard.2018.07.029). 

- We thank the reviewers for this important suggestion. The impact of psychological factors and other “confounders” of cognitive function in multiple sclerosis is now addressed.

  1. Similarly, the section on Neuropsychological Assessment should also mention the need for referrals to neuropsychologists not only to assess cognition, but also to provide a comprehensive formulation of patients' psychological state and difficulties. 

- The importance of screening and assessment of psychological state and difficulties has been highlighted in the present version

  1. In the Conclusions, the authors state that new forms of personalised and non-pharmacological interventions should be explored in the future. However, the review only mentions cognitive rehabilitation as non-pharmacological interventions, with no discussion around forms of psychological therapy which may help improve cognitive impairments in MS. This needs to be addressed, since psychological interventions such as mindfulness-based approaches have shown to be effective at improving cognitive functioning both in MS (https://doi.org/10.1080/09638288.2022.2069292) and comparable conditions such as premanifest/early stage Huntington's disease (https://doi.org/10.3233/JHD-210471).

- In the revised version we highlighted the role of psychological intervention such as mindfulness-base approaches. However, we did not include the citation of its efficacy in premanifest/early stage of HD, since it is not a comparable condition to MS.

Minor points: 

- On line 34 the author say "despite the historical description", without specifying what such description entailed. This could be added briefly.

- A brief description of the historical picture has been included.

- Starting on lines 52-53 it is not clear what the authors mean with "the main determinants of CI are greater physical disability..." - does this mean greater physical disability predicts higher CI or vice versa? The whole paragraph up until line 70 could use a comprehensive revision to improve clarity. It would also be helpful if the authors detailed what hypotheses/explanations have been proposed for CI to increase risk of death. 

- Sentences have been rephrased, for clarity

- The whole paper would benefit from being proofread by a native speaker of English in order to address inaccuracies and improve flow and readability. 

  • The paper received a linguistic revision.

Reviewer 3 Report

29 August 2022

Review on the manuscript titled ‘Cognitive impairment in multiple sclerosis: an update’ by Portaccio E & Amato MP, submitted to NeuroSci

Manuscript ID: neurosci-1900286 

Dear Authors, 

Cognitive impairment (CI) is a common presentation in patients with multiple sclerosis (MS). However, CI in MS remains underrecognized and thus, it is often unassessed and is left unmanaged. In the present review entitled ‘Cognitive impairment in multiple sclerosis: an update’, Portaccio and Amato reviewed recent evidence on prevalence, profile, pathogenetic mechanisms, and treatment of CI in patients with MS.

The main strength of this manuscript is that it addresses an interesting and timely topic, suggesting that more focused batteries of neuropsychological, laboratory, and radiological tests are necessary to build multimodal and personalized management for CI in patients with MS.

In general, I think the idea of this observational study is really interesting and the authors’ fascinating observations on this timely topic may be of interest to the readers of NeuroSci. However, some comments, as well as some crucial evidence that should be included to support the authors’ argumentation, needed to be addressed to improve the quality of the manuscript, its adequacy, and its readability prior to the publication in the present form. My overall opinion is to publish this research article after the authors have carefully considered my suggestions below, particularly reshaping parts of the Introduction section by adding more evidence.

Comments:

1.     Title: The title is too general. I suggest presenting more specific title representing this review and expert opinions of the authors.

2.     Abstract: Please clearly and proportionally present the background, the objectives, the summary, and the conclusion proportionally with 200 words.

3.     Keyword: Please list ten keywords and use as many keywords as possible in the first two sentences of the abstract.

4.     A Graphical Abstract is highly recommended.

5.     In general, I recommend authors to use more evidence to back their claims, especially in the Introduction of the article, which I believe is currently lacking. Thus, I recommend that the authors attempt to deepen the subject of their manuscript, as the bibliography is too concise. In my opinion, less than 150 references for a review research article are insufficient. Therefore, I suggest focusing their efforts on researching more relevant literature: I believe that adding more studies and reviews will help provide better and more accurate background to this review.

6.     Introduction: The ‘Introduction’ section is too concise and failed in providing enough information on MS in general. Even though the authors decided to take a narrow view of CI in MS, I believe that a deeper examination of the mechanisms underlying cognitive impairments including learning and memory, among others and how these abilities, together with deficient inhibitory control, are core factors in different neurodegenerative disorders, including MS, would provide a useful background. A recent review outlined typical dysfunctional behaviors, such as deficit in action control and motor inhibition, which are associated with psychopathological and neurologic conditions(https://doi.org/10.1016/j.brat.2021.103963). Another recent review focused on pathological mechanisms underlying altered emotion perception, which is significantly impaired in brain-damaged patients, are related to amygdala and superior temporal sulcus dysfunctions (https://doi.org/10.3390/biomedicines10030627).Thus, I recommend that the authors clarify CI, cognitive relapse, and isolated cognitive relapse in MS.

7.     Introduction: Finally, I also believe that a recent perspective manuscript on the metabolic pathways (https://doi.org/10.17219/acem/139572) involved in the pathogenesis of a wide range of diseases might be of interest. Moreover, it deserves to clarify that the current diagnostic subcategory of MS remains arbitrary, and it is difficult to exclude the possible presence of different pathogenesis. That said, the authors can also check additional studies that have focused on searching for biomarkers for redox status and mitochondrial resilience, developing precision diagnostics, and underlying processes causing impairments in social cognition and social functioning, associated with various psychiatric, neurological and neurodegenerative illnesses (https://doi.org/10.3390/neurolint14020030; https://doi.org/10.3390/life11121365; https://doi.org/10.3390/diagnostics12010130; https://doi.org/10.3390/biomedicines8100406; https://doi.org/10.3390/cells11162607; https://doi.org/10.3390/jpm11101032; https://doi.org/10.3390/biomedicines9040403). Having provided enough background, I recommend that the authors present the purpose of this review and summarize the sequence of the following sections.

8.     The sections 2-5: After the introduction, the authors here present detailed reviews in each section and describe important elements in depth.

9.     Discussion: After presenting each section, I recommend that the authors present the discussion section to fully expand arguments including the weakness, the limitations, the potential of this review, the goal, the challenge, the knowledge and the technology necessary to achieve this goal, and future research direction, among others.

10. Figures: I suggest presenting informative figures.

11. Conclusion: In this section I recommend that the authors present the take home message as the core part of this manuscript. Thus, please provide a synthesis of the data presented in the paper as well as possible keys to advance research and practice CI assessment in patients with MS.

12. References: Authors should consider revising the bibliography, as there are several incorrect citations. Indeed, according to the Journal’s guidelines, they should provide the abbreviated journal name in italics, the year of publication in bold, the volume number in italics for all the references (https://www.mdpi.com/journal/neurosci/instructions).

Overall, the manuscript contains no figure, one table and 54 references. I believe that this manuscript might carry important value reviewing prevalence, profile, pathogenetic mechanisms, and treatment of CI in patients with MS and concluding the need of diagnostic batteries for personalized management. I hope that, after these careful revisions, the manuscript can meet the Journal’s high standards for publication.

Best regards,

Reviewer

Author Response

  • We thank the reviewer for her/his comments and suggestions. However, the main objective of this review (it is not an observational/research study) was to provide a brief and concise overview of cognitive impairment. Despite this, in the revised version the manuscript has been expanded according to the reviewers’ suggestions.

Comments:

  1. Title: The title is too general. I suggest presenting more specific title representing this review and expert opinions of the authors.

  • The title has been modified including more specific information

  1. Abstract: Please clearly and proportionally present the background, the objectives, the summary, and the conclusion proportionally with 200 words.

  • The abstract is not structured as that of research article

  1. Keyword: Please list ten keywords and use as many keywords as possible in the first two sentences of the abstract.

  • Keywords have been expanded

  1. Graphical Abstractis highly recommended.

  • Since it is a review, we did not provide a graphical abstract

  1. In general, I recommend authors to use more evidence to back their claims, especially in the Introduction of the article, which I believe is currently lacking. Thus, I recommend that the authors attempt to deepen the subject of their manuscript, as the bibliography is too concise. In my opinion, less than 150 references for a review research article are insufficient. Therefore, I suggest focusing their efforts on researching more relevant literature: I believe that adding more studies and reviews will help provide better and more accurate background to this review.

  • As stated above, this is a brief review. A comprehensive and extensive literature review was beyond the scope of our manuscript. However, the content of the manuscript and bibliography have been expanded.

  1. Introduction: The ‘Introduction’ section is too concise and failed in providing enough information on MS in general. Even though the authors decided to take a narrow view of CI in MS, I believe that a deeper examination of the mechanisms underlying cognitive impairments including learning and memory, among others and how these abilities, together with deficient inhibitory control, are core factors in different neurodegenerative disorders, including MS, would provide a useful background. A recent review outlined typical dysfunctional behaviors, such as deficit in action control and motor inhibition, which are associated with psychopathological and neurologic conditions(https://doi.org/10.1016/j.brat.2021.103963). Another recent review focused on pathological mechanisms underlying altered emotion perception, which is significantly impaired in brain-damaged patients, are related to amygdala and superior temporal sulcus dysfunctions (https://doi.org/10.3390/biomedicines10030627).Thus, I recommend that the authors clarify CI, cognitive relapse, and isolated cognitive relapse in MS.

  • We limited the introduction of this brief review to MS-related background. While mechanisms underlying cognitive impairments and dysfunctional behaviors in other psychopathological and neurologic conditions are interesting and fascinating, they fall outside the scope of this review. CI, cognitive relapse, and isolated cognitive relapse in MS have been clarified in the paragraph 2.Prevalence and profile of cognitive impairment in multiple sclerosis.

  1. Introduction: Finally, I also believe that a recent perspective manuscript on the metabolic pathways (https://doi.org/10.17219/acem/139572) involved in the pathogenesis of a wide range of diseases might be of interest. Moreover, it deserves to clarify that the current diagnostic subcategory of MS remains arbitrary, and it is difficult to exclude the possible presence of different pathogenesis. That said, the authors can also check additional studies that have focused on searching for biomarkers for redox status and mitochondrial resilience, developing precision diagnostics, and underlying processes causing impairments in social cognition and social functioning, associated with various psychiatric, neurological and neurodegenerative illnesses (https://doi.org/10.3390/neurolint14020030 https://doi.org/10.3390/life11121365; https://doi.org/10.3390/diagnostics12010130 ); https://doi.org/10.3390/biomedicines8100406; https://doi.org/10.3390/cells11162607; https://doi.org/10.3390/jpm11101032 (https://doi.org/10.3390/biomedicines9040403 (. Having provided enough background, I recommend that the authors present the purpose of this review and summarize the sequence of the following sections.

  • As stated above, the description of pathogenetic mechanisms involved in other diseases falls outside the scope of this review. The alterations of metabolic pathways, redox status and mitochondrial resilience are fascinating, but relevant in a more extensive and comprehensive literature review on the field. Moreover, it is not clear to me what the reviewer referred stating that “the current diagnostic subcategory of MS remains arbitrary”. Finally, the great majority of suggested references are not related to the objective of the present review.

  1. The sections 2-5: After the introduction, the authors here present detailed reviews in each section and describe important elements in depth.

  1. Discussion: After presenting each section, I recommend that the authors present the discussion section to fully expand arguments including the weakness, the limitations, the potential of this review, the goal, the challenge, the knowledge and the technology necessary to achieve this goal, and future research direction, among others.

  • The main limitation of this revision has been already clearly stated in the introduction, that is its conciseness.

  1. Figures: I suggest presenting informative figures.

  • In this concise review we limited to one single table

  1. Conclusion: In this section I recommend that the authors present the take home message as the core part of this manuscript. Thus, please provide a synthesis of the data presented in the paper as well as possible keys to advance research and practice CI assessment in patients with MS.

  • The conclusion provides a synthesis of data presented and take-home messages.

  1. References:Authors should consider revising the bibliography, as there are several incorrect citations. Indeed, according to the Journal’s guidelines, they should provide the abbreviated journal name in italics, the year of publication in bold, the volume number in italics for all the references (https://www.mdpi.com/journal/neurosci/instructions).

  • References have been revised, according to the reviewer’s suggestions.

Overall, the manuscript contains no figure, one table and 54 references. I believe that this manuscript might carry important value reviewing prevalence, profile, pathogenetic mechanisms, and treatment of CI in patients with MS and concluding the need of diagnostic batteries for personalized management. I hope that, after these careful revisions, the manuscript can meet the Journal’s high standards for publication.

- Again, we underscore that the present manuscript is a brief and concise overview of cognitive impairment in MS. We thank the reviewer for the insightful suggestions that would deserve a more extensive and comprehensive literature revision, falling outside the scope of our paper.

Round 2

Reviewer 1 Report

The manuscript is updated now and more comprehensive. Thank you. 

Author Response

Thank you again for your insightful revision

Reviewer 2 Report

The authors have addressed my concerns satisfactorily, I am happy to accept the manuscript in its present form. 

Author Response

(The authors gave the same response as above.)

Reviewer 3 Report

 The authors partially revised the manuscript and most of my message has not been conveyed in the second version of this review article. I believe that the manuscript will benefit from careful consideration suggested in this review report eventually to meet the high standard of the journal for publication. This second round review report iterates several points which have not been carefully considered and thus remain unrefined.

Comments:

1.     Abstract: Please expand the abstract to 200 words. I suggest, as a review article, proportionally presenting the background (general, detailed, and current issues addressed to this review), the objectives, the summary, and the conclusion

2.     In general, I recommend authors to use more evidence to back their claims, especially in the Introduction of the article, which I believe is currently lacking. Thus, I recommend that the authors attempt to deepen the subject of their manuscript, as the bibliography is too concise. In my opinion, less than 150 references for a review research article are insufficient. Therefore, I suggest focusing their efforts on researching more relevant literature: I believe that adding more studies and reviews will help provide better and more accurate background to this review.

3.     Introduction: This section is too concise and failed in providing enough information on multiple sclerosis (MS) in general. Even though the authors decided to take a narrow view of cognitive impairment (CI) in MS, I would like to emphasize here that this manuscript will greatly benefit from taking a broader view in the beginning to reinforce its comprehensiveness, to present its relevance to many readers, and thus to make it able to stand out the importance of CI in MS, which is a core topic of this review. A deeper examination of the mechanisms underlying CI including learning and memory, among others and how these abilities, together with deficient inhibitory control, are core factors in different neurodegenerative disorders, including MS, would provide a useful background. A recent review outlined typical dysfunctional behaviors, such as deficit in action control and motor inhibition, which are associated with psychopathological and neurologic conditions(https://doi.org/10.1016/j.brat.2021.103963). Another recent review focused on pathological mechanisms underlying altered emotion perception, which is significantly impaired in brain-damaged patients, are related to amygdala and superior temporal sulcus dysfunctions (https://doi.org/10.3390/biomedicines10030627).Thus, I recommend that the authors clarify CI, cognitive relapse, and isolated cognitive relapse in MS.

4.     Finally, I also believe that a recent perspective manuscript on the metabolic pathways (https://doi.org/10.17219/acem/139572) involved in the pathogenesis of a wide range of diseases might be of interest. Moreover, it deserves to clarify that the current diagnostic subcategory of MS remains not always useful for treatment decision, and it is difficult to exclude the possible presence of different pathogenesis. That said, the authors can also check additional studies that have focused on searching for biomarkers for redox status and mitochondrial resilience, developing precision diagnostics, and underlying processes causing impairments in social cognition and social functioning, associated with various psychiatric, neurological and neurodegenerative illnesses (https://doi.org/10.3390/neurolint14020030; https://doi.org/10.3390/life11121365; https://doi.org/10.3390/diagnostics12010130; https://doi.org/10.3390/biomedicines8100406; https://doi.org/10.3390/cells11162607; https://doi.org/10.3390/jpm11101032; https://doi.org/10.3390/biomedicines9040403). Having provided enough background, I recommend that the authors present the purpose of this review and summarize the sequence of the following sections.

5.     Figures: I suggest presenting informative figures.

6.     Conclusion: In this section, please clarify the synthesis of the data presented in this review as well as possible keys to advance research and practice CI assessment in patients with MS. I recommend that the authors present the take home message as the core part of this manuscript.

7.     References: The reference style remain incorrect such as the number of authors, et al., missing period after journal abbreviation, unnecessary period after journal names, and unnecessary date of publication, among others. Please carefully have a look and correct it according to the journals’ guidelines (https://www.mdpi.com/journal/neurosci/instructions).

Overall, the manuscript contains no figure, one table and 68 references. 

Author Response

The authors partially revised the manuscript and most of my message has not been conveyed in the second version of this review article. I believe that the manuscript will benefit from careful consideration suggested in this review report eventually to meet the high standard of the journal for publication. This second round review report iterates several points which have not been carefully considered and thus remain unrefined.

Comments:

  1. Abstract: Please expand the abstract to 200 words. I suggest, as a review article, proportionally presenting the background (general, detailed, and current issues addressed to this review), the objectives, the summary, and the conclusion

- The abstract has been expanded and restructured, as suggested.

  1. In general, I recommend authors to use more evidence to back their claims, especially in the Introduction of the article, which I believe is currently lacking. Thus, I recommend that the authors attempt to deepen the subject of their manuscript, as the bibliography is too concise. In my opinion, less than 150 references for a review research article are insufficient. Therefore, I suggest focusing their efforts on researching more relevant literature: I believe that adding more studies and reviews will help provide better and more accurate background to this review.

  • In the present version we deepened the subject of the manuscript, in particular by adding reference to different potential mechanisms underlying cognitive impairment in MS (see below). The number of references in the bibliography now raised to 74. We believe that this is sufficient for a brief, concise review of this topic.

  1. Introduction: This section is too concise and failed in providing enough information on multiple sclerosis (MS) in general. Even though the authors decided to take a narrow view of cognitive impairment (CI) in MS, I would like to emphasize here that this manuscript will greatly benefit from taking a broader view in the beginning to reinforce its comprehensiveness, to present its relevance to many readers, and thus to make it able to stand out the importance of CI in MS, which is a core topic of this review.A deeper examination of the mechanisms underlying CI including learning and memory, among others and how these abilities, together with deficient inhibitory control, are core factors in different neurodegenerative disorders, including MS, would provide a useful background. A recent review outlined typical dysfunctional behaviors, such as deficit in action control and motor inhibition, which are associated with psychopathological and neurologic conditions(https://doi.org/10.1016/j.brat.2021.103963). Another recent review focused on pathological mechanisms underlying altered emotion perception, which is significantly impaired in brain-damaged patients, are related to amygdala and superior temporal sulcus dysfunctions (https://doi.org/10.3390/biomedicines10030627).Thus, I recommend that the authors clarify CI, cognitive relapse, and isolated cognitive relapse in MS.

  • Examination of the mechanisms underlying cognitive impairment has been expanded, including reference to deficient inhibitory control and altered emotion perception. Cognitive relapse and isolated cognitive relapse are described on page 2, lines 54-56.

  1. Finally, I also believe that a recent perspective manuscript on the metabolic pathways (https://doi.org/10.17219/acem/139572) involved in the pathogenesis of a wide range of diseases might be of interest. Moreover, it deserves to clarify that the current diagnostic subcategory of MS remains not always useful for treatment decision, and it is difficult to exclude the possible presence of different pathogenesis. That said, the authors can also check additional studies that have focused on searching for biomarkers for redox status and mitochondrial resilience, developing precision diagnostics, and underlying processes causing impairments in social cognition and social functioning, associated with various psychiatric, neurological and neurodegenerative illnesses (https://doi.org/10.3390/neurolint14020030 ; https://doi.org/10.3390/life11121365 ; https://doi.org/10.3390/diagnostics12010130 ; https://doi.org/10.3390/biomedicines8100406; https://doi.org/10.3390/cells11162607; https://doi.org/10.3390/jpm11101032 ; https://doi.org/10.3390/biomedicines9040403). Having provided enough background, I recommend that the authors present the purpose of this review and summarize the sequence of the following sections.

  • Again, I do not understand what the reviewer means stating that “the current diagnostic subcategory of MS remains not always useful for treatment decision, and it is difficult to exclude the possible presence of different pathogenesis” and its relation with cognitive impairment in MS. Which are the diagnostic subcategories? Is she/he referring to disease modifying treatment or symptomatic treatment? As for pathogenetic mechanisms of cognitive impairment involved in other diseases, in the present version we discussed alterations of metabolic pathways, redox status and mitochondrial resilience. Data on these mechanisms in MS are scant, but have been anyway included. It has to be noted again, however, that the majority of suggested references are not related to the objective of the present review (for instance https://doi.org/10.3390/neurolint14020030 is about COVID19 and MS; https://doi.org/10.3390/life11121365 is about statin treatment; https://doi.org/10.3390/jpm11101032 is about genetic and response to treatment with glatiramer acetate)

  1. Figures: I suggest presenting informative figures.

  • A figure has been provided

  1. Conclusion: In this section, please clarify the synthesis of the data presented in this review as well as possible keys to advance research and practice CI assessment in patients with MS. I recommend that the authors present the take home message as the core part of this manuscript.

  • Take home messages have been added in a dedicated Table

  1. References: The reference style remain incorrect such as the number of authors, et al., missing period after journal abbreviation, unnecessary period after journal names, and unnecessary date of publication, among others. Please carefully have a look and correct it according to the journals’ guidelines (https://www.mdpi.com/journal/neurosci/instructions).

.-    References have been revised

Overall, the manuscript contains no figure, one table and 68 references. 

In the revised version, the manuscript contains one figure, two tables, 74 references.

Round 3

Reviewer 3 Report

The 2nd review on the manuscript titled ‘Cognitive impairment in multiple sclerosis: an update’ by Portaccio E & Amato MP, submitted to NeuroSci

Manuscript ID: neurosci-1900286 

Dear Authors, 

The authors partially revised the manuscript and most of my message has not been conveyed in the second version of this review article. I believe that the manuscript will benefit from careful consideration suggested in this review report eventually and hopefully to meet the high standard of the journal for publication. This second round review report iterates several points which have not been carefully considered and thus, many parts remain unrefined.

Comments:

1.     Title: Adding two words could hardly improve the title. I recommend presenting the title stating the importance of this review.

2.     Abstract: Please expand the abstract to 200 words. I suggest, as a review article, proportionally presenting the background, the objectives, the summary, and the conclusion.  The background should describe general, detailed, and current issues addressed to this review. The conclusion should state the potential and the advance this review has provided in the field.

3.     The review article benefits a lot from a graphical abstract, likewise to original articles.

4.     In general, I recommend authors to use more evidence to back their claims, especially in the Introduction of the article, which I believe is currently lacking. Thus, I recommend that the authors attempt to deepen the subject of their manuscript, as the bibliography is too concise. As it stands, 68 references could hardly provide basis and evidence in arguments the authors would develop. In my opinion, less than 150 references for a review research article are insufficient. Therefore, I suggest focusing their efforts on researching more relevant literature: I believe that adding more studies and reviews will help provide better and more accurate background to this review.

5.     Introduction: This section is too concise and failed in providing enough information on multiple sclerosis (MS) in general. Even though the authors decided to take a narrow view of cognitive impairment (CI) in MS, I would like to emphasize here that this manuscript will greatly benefit from taking a broader view in the beginning to reinforce its comprehensiveness, to present its relevance to many readers, and thus to make it able to stand out the importance of CI in MS, which is a core topic of this review. A deeper examination of the mechanisms underlying CI including learning and memory, among others and how these abilities, together with deficient inhibitory control, are core factors in different neurodegenerative disorders, including MS, would provide a useful background. A recent review outlined typical dysfunctional behaviors, such as deficit in action control and motor inhibition, which are associated with psychopathological and neurologic conditions(https://doi.org/10.1016/j.brat.2021.103963). Another recent review focused on pathological mechanisms underlying altered emotion perception, which is significantly impaired in brain-damaged patients, are related to amygdala and superior temporal sulcus dysfunctions (https://doi.org/10.3390/biomedicines10030627).Thus, I recommend that the authors clarify CI, cognitive relapse, and isolated cognitive relapse in MS.

6.     Finally, I also believe that a recent perspective manuscript on the metabolic pathways (https://doi.org/10.17219/acem/139572) involved in the pathogenesis of a wide range of diseases might be of interest. Moreover, it deserves to clarify that the current diagnostic subcategory of MS remains not always useful for treatment decision, and it is difficult to exclude the possible presence of different pathogenesis. That said, the authors can also check additional studies that have focused on searching for biomarkers for redox status and mitochondrial resilience, developing precision diagnostics, and underlying processes causing impairments in social cognition and social functioning, associated with various psychiatric, neurological and neurodegenerative illnesses (https://doi.org/10.3390/neurolint14020030; https://doi.org/10.3390/life11121365; https://doi.org/10.3390/diagnostics12010130; https://doi.org/10.3390/biomedicines8100406; https://doi.org/10.3390/cells11162607; https://doi.org/10.3390/jpm11101032; https://doi.org/10.3390/biomedicines9040403). Having provided enough background, I recommend that the authors present the purpose of this review and summarize the sequence of the following sections.

7.     Figures: I suggest presenting informative figures.

8.     Conclusion: Adding one sentence could hardly do its work, as I suggested below. In this section, please clarify the synthesis of the data presented in this review as well as possible keys to advance research and to practice CI assessment in patients with MS. I recommend that the authors present the take home message as the core part of this manuscript.

9.     References: First, please follow the citation style according to the journal’s guidelines.  The number of citations should be in the brackets like this [1]. The reference style remains incorrect such as the number of authors, et al., missing period after journal abbreviation, unnecessary period after journal names, and unnecessary date of publication, among others. Please carefully have a look and correct it according to the journals’ guidelines (https://www.mdpi.com/journal/neurosci/instructions).

Overall, the manuscript contains no figure, one table and 68 references. I believe that this manuscript might carry important value reviewing prevalence, profile, pathogenetic mechanisms, and treatment of CI in patients with MS and concluding the need of diagnostic batteries for personalized management. I hope that, after these careful revisions, the manuscript can meet the Journal’s high standards for publication.

Best regards,

Reviewer